
# Numerical modeling of the 2013 meteorite entry in Chebarkul Lake, Russia

Andrey Kozelkov [1,2], Andrey Kurkin[2], Efim Pelinovsky[2,3], Vadim Kurulin [1], Elena Tyatyushkina[1]

[1]Russian Federal Nuclear Center, All-Russian Research Institute of Experimental Physics, Sarov, 607189, Russia
[2]Nizhny Novgorod State Technical University n.a. R.E. Alekseev, Nizhny Novgorod, 603950, Russia
[3]Institute of Applied Physics, Nizhny Novgorod, 603950, Russia

*Correspondence to*: Andrey Kurkin (aakurkin@gmail.com)

**Abstract.** The results of the numerical simulation of possible hydrodynamic perturbations in Lake Chebarkul (Russia) as a consequence of the meteorite fall of 2013 (Feb. 15) are presented. The numerical modeling is based on the Navier-Stokes equations for a two-phase fluid. The results of the simulation of a meteorite entering the water at an angle of 20 degrees are given. Numerical experiments are carried out both when the lake is covered with ice and when it isn't. The estimation of size of the destructed ice cover is made. It is shown that the size of the observed ice-hole at the place of the meteorite fall is in good agreement with the theoretical predictions, as well as with other estimates. The heights of tsunami waves generated by a small meteorite entering the lake are small enough (a few centimeters) according to the estimations. However, the danger of a tsunami of meteorite or asteroid origin should not be underestimated.

## 1 Introduction

February 15, 2013 at 9:20 a.m. local time, in the vicinity of the city of Chelyabinsk, Russia (Fig. 1) a meteorite exploded and collapsed in the earth's atmosphere as a result of inhibition. Its fall was accompanied by a series of atmospheric explosions and propagation of shock waves over the territory of the Chelyabinsk region. Small fragments of the meteorite came down on the region. It is the largest of the known celestial bodies which fell to the ground after the Tunguska meteorite in 1908. The following characteristics of the Chelyabinsk meteorite are given in recent publications [Zetser, 2014, Ionov, 2013, Kopeikin et al., 2013, Emel'yanenko et al., 2013, Popova et al., 2013, Berngardt et al., 2013, Gokhberg et al., 2013, Krasnov et al., 2014, Seleznev et al., 2013, De Groot-Hedlin and Hedlin, 2014]:

- the meteorite with the diameter of 16-19 meters flew into the earth's atmosphere at about 20 degrees to the horizon with the velocity ~ 17-22 km/s;
- the meteorite was destroyed in several stages, and the main explosion occurred at an altitude of about 23 km. The analysis of the meteorite fall led to the estimation of the explosion energy from 380 to 1,000 kilotons of TNT;
- surviving fragments of the Chelyabinsk meteorite supposedly flew at velocities up to 150 - 300 m/s, and according to the updated data, the velocity was 156 m/s;



- the largest meteorite fragment, weighing about 550 kg, was recovered from Lake Chebarkul near the Krutik Peninsula and it had an irregular oval shape with the average outer diameter of about 1 m;

- the celestial body was exploded when it hit the ice and water. Its fragments flew more than 100 m and formed an ice-hole of the round shape with the diameter of about 8 m (Fig. 1).

The GPR survey of the crash site showed that the crater, which was formed by the meteorite impact on the bottom, was observed 30 m away from the ice-hole [Kopeikin et al., 2013].

The results of the numerical modeling of the processes that began in the water after the meteorite entering Lake Chebarkul are presented in this paper. We consider two cases – the occurrence of a meteorite in the ice-covered lake (as it was in February 2013), and in the lake without ice to estimate the amplitude of the wave in the case of a possible fall in the summer.

First of all, in section 2 is used a well-known parametric model of the tsunami source of meteoric origin [Ward and Asphaug, 2000] which estimates the disturbances in the water at the site of the meteorite entry. It is shown that the size of the source is twice the size of the observed ice-hole; that is why this model cannot explain the observed phenomenon. In section 3 we describe the hydrodynamic model of wave generation based on the Navier-Stokes equations as well as the parameters of its numerical discretization. In section 4 we present the results of numerical experiments on the tsunami wave generation in

open water. Also is made the estimation of the wave heights, the description of the stages of the meteorite collapse and as well as the generation of waves emanating from the source. The results of numerical experiments on disturbance generation in the lake covered with ice are presented in section 5. The zones of maximum and minimum pressure, which could potentially lead to the destruction of the ice cover are detected. In section 6 are shown the results of estimations of the ice cover destruction, based on the calculation of the stresses. The area of ice destruction is estimated according to the famous

semi-empirical formulas and pressure values obtained in the numerical experiments. This estimation is in good agreement with the observed data, as well as with the estimates made earlier in [Ivanov, 2014]. The results are summarized in section 7.

## 2 Preliminary estimates of wave heights in the lake with a free surface

To estimate the possible parameters of the water surface displacement when a meteorite falls, we use a simplified model in which the generation of waves by a meteorite entering the water is parameterized by certain initial conditions [Ward and

Asphaug, 2000; Mirchina and Pelinovsky, 1988; Kharif and Pelinovsky, 2005; Levin and Nosov, 2009; Torsvick et al, 2010]. It is suggested that in the initial stage of a crater formation a meteorite, vertically entering the water, creates a radially symmetrical cavity on the water surface, which can be described by a simple function:

$$\eta^{imp}(r) = D_C(1 - r^2 / R_C^2), \ r \le R_D,$$
$$\eta^{imp}(r) = 0, \qquad\qquad r > R_D, \tag{1}$$



where $D_c$ is the depth of the cavity, $R_c$ and $R_D$ are inner and outer radii of the cavity, respectively. In the case of $R_D = \sqrt{2}R_C$ the water ejected from the cavity, forms the outer splash – ring structure typical of the fall of the object in the water and the volume of which corresponds exactly to the volume of water discharged from the cavity.

Considering that the meteorite kinetic energy is converted into the potential energy of the water level displacement, the following simple analytical formulas are derived to calculate the radius and depth of the cavity; see for instance, [Ward and Asphaug, 2000]:

$$D_C = \sqrt{\frac{2\varepsilon\rho_I R_I^3 V_I^2}{\rho_w g R_C^2}}, \quad R_C = R_I \left( 2\varepsilon \frac{V_I^2}{gR_I} \right)^{\delta} \left( \frac{\rho_I}{\rho_w} \right)^{1/3} \left( \left( \frac{\rho_w}{\rho_I} \right)^{1/3 - \delta} \left( \frac{1}{qR_I^{\alpha-1}} \right)^{2\delta} \right), \tag{2}$$

where $\rho_w$ is density of water, $g$ is gravitational acceleration, $\varepsilon$ is the proportion of the kinetic energy of the meteorite, converted into the tsunami energy, $\rho_I$, $R_I$, $V_I$ are density, radius and velocity of the meteorite $q$ and $\alpha$ are the coefficients, associated with the properties of the meteorite and the water layer. According to [Levin and Nosov, 2009] about 16% of the kinetic energy of the falling body is converted into the energy of tsunami waves. The parameter $\alpha$ is 1.27 and the value of $q$ and $\delta$ is calculated as follows:

$$\delta = \frac{1}{2\alpha + 2}, \quad q \approx 0,39 \left( \frac{\rho_w}{\rho_I} \right)^{0,26} \frac{1}{R_I^{0,27}}. \tag{3}$$

If the diameter of a meteorite is 1 meter, the density is 3.3 g/cm$^3$ and the water entering velocity is 156 m/s, the inner radius of the cavity is equal to $R_C = 5.6$ m, the outer radius is $R_D = 7.8$ m and the depth of the cavity formed on the surface of Lake Chebarkul is $D_C = 3.2$ m approximately. The resulting cavity is schematically depicted in Fig. 2.

The obtained estimation of the cavity diameter (15.5 m) exceeds twice the size of the ice-hole (7-8 m) formed by the fall of a Chelyabinsk meteorite fragment (Fig. 1). The estimation of the cavity size is made for the conditions of open water. If we take into account the ice cover, which prevents the water from forming vertical upward discharges at the edges of the resulting cavities, and consider only part of the underwater cavity (indicated in Fig. 2 by dashed lines), we will obtain the diameter of a little more than 10 meters, which is closer to the observed size of the ice-hole. Another reason for the differences in the estimation of the source size, in our opinion, might be the wrong extrapolation of empirical formulas for the small size meteorites, especially since it does not enter the water vertically.

## 3 The hydrodynamic model of the tsunami source, based on the Navier-Stokes equations

The models based on the numerical solution of the Navier-Stokes equations [Ferziger and Peric, 2002], which have become popular in solving the problems of asteroid tsunami, allow simulating a real meteorite entering the water [Kozelkov et al., 2015a, 2016a]. In this paper we solve the Navier-Stokes equations for a two-component (water and air) incompressible fluid in a gravitational field, and both air and water are considered to be incompressible. These equations are used to describe each





component of the fluid. Their integration into a single system is made by using the method of «Volume of Fluid» (VOF) [Hirt and Nichols, 1981]. The numerical solution of the resulting system is made by using the methods SIMPLE/PISO [Ferziger and Peric, 2002, Issa, 1986]. The movement of the meteorite is modeled by using the method of immersed boundary (Immersed Boundary Method, IBM) [Mittal and Iaccarino, 2005]. This method involves the allocation of cells

completely or partially occupied by a rigid body in the computational field and resistance force, proposed in [Mohd-Yusof, 1997]. This approach to modeling the moving rigid body is quite simple. It does not require dynamic rebuilding of the computational grid thus giving good results for the practical tasks that do not need a detailed description of the boundary layer near the surface of a solid body [Posa et al., 2011].

In the present calculations are used zero initial conditions for the velocity field in the water, the water level and the standard

hydrostatic distribution of the pressure. At the boundaries and at the bottom of the computational domain (distant from the site of a meteorite entering the water), the boundary condition «rigid wall» is used. It assumes a zero velocity and zero value of the pressure gradient and the volume fraction of the components, as well as the zero shear stress. In fact, the calculation stops when the perturbation reaches the lateral boundaries so that these boundaries do not actually affect the results. The upper part of the computational domain is an open border with the given static pressure of 1 atm, and zero velocity gradient.

The bottom of the basin is regarded deformable, and when a rigid body reaches the bottom, its velocity is artificially set to zero. The value of the component volume fraction at the upper boundary is different and depends on the direction of the flow: the inlet flow volume fraction of water is equal to zero and of the air is equal to one. At the output flow the gradient of the volume fraction of each component is assumed to be zero. Physically, this condition means that all the components are free to leave the settlement area through the upper boundary, but only the air enters it.

The described model is implemented in the software package LOGOS, which is used for the numerical simulations presented below. The LOGOS software package has been tested for the given class of problems and has shown sufficiently reliable results [Kozelkov et al., 2013, Kozelkov et al., 2014b, 2015b, 2016b, Volkov et al., 2013, Betelin et al., 2014, Deryugin et al., 2015].

The bathymetric data of Lake Chebarkul are not available, and the depth of a model region relative to the surface is taken

constant of 10 m, corresponding to an average depth of Lake Chebarkul. The height above the zero level surface of the water is 40 m. So, the model discrete area with the size of 160×160×50 m was constructed. It is a non-structured three-dimensional (3D) grid of truncated polyhedrons of arbitrary shape (Fig. 3). This type of grid is the only possible one for the areas of complex geometric configurations. It is built by the preprocessor of the software package LOGOS.

The upper right part of Fig. 3 displays the structure of the vertical grid coordinate. The distance from the surface of the water

(zero) to the maximum value of the upper boundary of the air is 40 m and the distance from the surface to the bottom is 10 m. Typical cell dimensions in the air are from 0 to 0.2 m – $\Delta z = 10$ cm; from 0.2 m to 15.5 m – $\Delta z = 25$ cm; from 15.5 m to 17 m – $\Delta z = 50$ cm; from 17 m to 21 m – $\Delta z = 0.75$ m; from 21 m to 27 m – $\Delta z = 1.5$ m; and from 27 m to 35 m – $\Delta z = 3$ m. The dimensions of cells in the water are (upright) from 0 to 0.2 m – $\Delta z = 10$ cm, from 0.2 m to 10 m – $\Delta z = 25$ cm. The



condensation of grid points in the cylindrical region to the typical cell size of 10 cm (while the meteorite diameter is at least 10 cells) is made to model the meteorite motion in the water correctly. This area is located from a slope of 20 degrees in accordance with the angle of the meteorite fall (the lower part of Fig. 3, right).

## 4 Numerical experiments on the generation of tsunami waves in open water

In the first numerical experiment we examine the case of wave generation in the water without ice cover. This simulation does not require the application of the models describing ice, and it can be held entirely within the hydrodynamic model described above. Fig. 4 illustrates a three-dimensional distribution of the water when the meteorite enters the water at an angle of 20 degrees to the vertical. The diameter of the meteorite is 1 m, its density is 3.3 g/cm$^3$ and the velocity of the entry is 156 m/s. The visualization is carried out by the post-processing software package LOGOS.

A powerful upward surge of water is formed at the first moments of the collision (the formation of the sultan), and then waves are generated. The head wave has a positive polarity (the hump). Under the water a strong air bubble is generated by the movement of the body in the water. At first this bubble has a cylindrical structure, and then, during the destruction and surfacing it becomes oval. The destruction of the underwater bubble near the surface generates an additional "secondary" wave. This process is in good agreement with the classic description of a body falling into the water [Aristoff, 2009].

The anisotropy of wave propagation is visible – the waves propagate faster in the direction of a meteorite fall than in the opposite direction, and their amplitudes decrease rapidly. The details of the cross-section of perturbations are shown in Figs. 5 and 6. Fig. 5 shows the surface of the water in the cross-section taken along the center line of the meteorite fall on a large scale, and Fig. 6 shows it in the whole computational domain.

The meteorite passes about a quarter of the water column in 0.05 seconds and it almost reaches the bottom in 0.15 seconds.

In 0.25 seconds the meteorite is already lying on the bottom, so it has covered a distance of about 27 meters from the ice-hole on the surface. This result is in good agreement with the detailed analysis of geo-radar cross-section survey [Kopeikin et al., 2013]. When the meteorite falls it forms a cavity with the diameter of about 4 meters near the surface in 0.05 seconds. The diameter becomes 7-8 meters before the start of the collapse, which begins when the meteorite reaches the bottom. This value corresponds to the diameter of the ice-hole, formed as a result of a real meteorite fall. The collapse of the cavity begins

0.25 seconds later and ends in 0.5 seconds. At the time of the cavity collapse are generated a "primary wave" with the height of about 3.5 meters and large air pocket in the water column. The height of the "core" wave corresponds to the theoretical estimation (2) in the framework of the parametric source. The height of "splash" lifting at the time of the cavity collapse is close to the mark of 10 meters, but these "splashes" do not participate in the surface wave generation (see. time 1-2 seconds). The transformation of the "air" pocket starts in 0.5 seconds. At the time of 2 seconds it reaches the water surface and

generates a "secondary" wave about 1 meter high. 2 seconds after a meteorite entering the water the height of the primary wave is about 1 m, it completely collapses and begins to move away from the source. At this time, is generated the "secondary" wave of almost the same height as the primary one. The process of the collapse ends 3 seconds after the meteorite fall. The maximum height of the sultan, formed after the fall, is observed approximately 0.75 seconds later. It is





about 10 meters, which corresponds to the depth of the basin (Fig. 5). This height is visible at the withering "cap" of the sultan, which breaks away from the main water discharge of about 4 m.

At the time of 1 second the first wave, which reaches about 1 meter high the time of 1.5 seconds, begins to propagate from the source. In 2 seconds this wave collapses under the air pocket impact, which reaches the water surface and generates the so-called "secondary" wave. The "primary" and "secondary" waves collide with each other in our numerical experiment. The maximum height of the generated wave is observed at the time of 3 seconds and it is about 3 m high. During the next second the water near the site of the meteorite fall oscillates, and 5 seconds after the meteorite enters the water two waves go out from the source. They could be clearly seen in Fig. 6 at a time of 6 seconds - their amplitude is only about 20 cm. These two leading waves go out from the source and then propagate on Lake Chebarkul. Generally speaking, there should be two waves formed by the meteorite – a "primary" wave from the intermediate cavity and a "secondary" wave generated by the collapse of the cavity and by the underwater bubble [Aristoff, 2009, Kozelkov, 2015]. However, here the waves had an impact on each other, and created a single wave, coming from the source. Subsequently, this wave is attenuated due to the cylindrical divergence and dispersion by law $r^{-(0.5-1)}$, therefore, its height at a distance greater than 100 meters will not exceed a few centimeters, that is why such a wave is not dangerous outside the area of the meteorite entering the water.

In conclusion we'd like to point out that the difference between the sizes of the ice-hole, obtained by using the parametric model and within the Navier-Stokes equations, may be concerned with the fact that the model does not take into account the angle of the fall. According to Fig. 6 (the time of 0.2 seconds – the meteorite immersion is total) the major discharge of water falls on the side of the cavity, which is located along the moving body. Its diameter is just 7-8 meters. If the fall had been vertical, the cavity would have looked different. Fig. 7 illustrates the cavities resulting from the different falls when the meteorite hits the bottom.

One can see that the geometry of the cavities varies and two intense waves are generated if the fall is vertical. In the fall at an angle a wave is formed along the meteorite moving, while behind it the disturbances are minimal and they will go in this direction just after the collapse of the "left" side.

As it was already noted, the bathymetric map of Lake Chebarkul is absent. It is possible to view tsunami wave propagation on the lake when the coastline is digitized and constant average depth taken (for lakes Chebarkul it is 10 meters).

Fig. 8 (left) shows the map of Lake Chebarkul shoreline and its digitized version on public maps built with grid model (right). As it can be seen, the coastline has a complicated configuration, and it is advisable to construct a grid model with the help of an automatic generator of arbitrary unstructured grids.

The grid constructed by this generator is shown in Fig. 8 (left). In the area free from the shoreline (open water) the grid has is mainly a hexagonal structure. Lake dimensions in length and width are approximately the same and make about 2 km. With the meteorite diameter of 1 meter it is reasonable to allocate a fall area to grind the grid model to the desired level (Fig. 8, right) – at least 10 cells along the meteorite diameter (typical cell size of 10 cm).

It is obvious that in case of building a grid for the entire lake with the characteristic size, its size will be very large, which will significantly affect the account time and the used CPU box.





A typical cell size outside the fall area can be assumed optional, here it ranges from 10 to 50 meters in the horizontal plane.

To model the wave propagation accurately it is necessary to thicken the grid near the interface of air and water surface (Fig. 8, top right). Usually, this thickening is carried out by the given law of geometric progression, so the typical cell sizes near the surface can be selected of the desired size in order to track the desired wave height. The size of the grid represented is

approximately 15 million cells. The results of tsunami propagation simulation on Lake Chebarkul without ice cover are shown in Fig. 9.

Already at the time of 9 seconds, it can be seen that the wave has sufficiently small amplitude of about 10 cm. About 30 seconds after the fall the wave reaches the nearest shore, and its height is also about 10 cm.

At the time of 45 seconds can be clearly seen the bounce of the wave from the shore. This, in its turn, suggests a fairly good

grid resolution which allows to reproducing centimeter amplitude waves.

At the time of 60 seconds, it can be seen how the bounced wave follows the primary one, which arose after the collapse of the water crater, formed in the course of a meteorite fall. 90 seconds after the fall, the primary wave as well as the "catch-up way" are practically damped. The wave does not reach the opposite bank.

## 5 Numerical experiments taking into account ice

The Chelyabinsk meteorite fell in winter time when the ice thickness on Lake Chebarkul was 70-80 cm. To model the meteorite fall into the water with punching the ice is quite a challenge. We have to take into account the relationship of processes of hydrodynamics and destruction. Therefore, here to estimate the destruction of the ice cover and the possible splash zone of the water on the ice surface we simulate an idealized case of a meteorite fall. Let a meteorite enter an already "punched" hole in the ice surface. In this case the ice is simulated as the boundary condition "rigid wall", and the zero water

level is placed in the hole. The hole in the computational experiment is rectangle with the size of 160x130 cm$^2$, close to the size of the meteorite. The meteorite entering parameters are exactly the same as in the previous case. Fig. 8 shows the results of the computational experiment.

The height of the sultan formed by the fall of the meteorite into the hole is over ten meters at the time of 0.2 seconds. Thus the sultan rises almost vertically upwards, without any deviation in any direction. The main part of the sultan consists of

splashes, and the height of the discharge of the water bulk is about 2.5 meters.

In about one second after the fall the sultan enters the final stage of the collapse on the ice surface. When the sultan collapses, it begins to deviate in the direction towards the entering body (left), without generating strong disturbances on the ice surface and under it. The maximum height of the wave falling on the ice surface is 1 m at the time of 1 sec. The collapsing sultan spans the distance of about 10 m on the ice surface in the direction of the fall (Fig. 10 at time 2 seconds).

The resulting size of the ice-hole during the Chelyabinsk event was several times larger than the diameter of the meteorite, i.e., the observed destruction of the ice cover around the area of the fall was much larger than the area of the collision. To analyze this phenomenon it is necessary to examine the pressure distribution in the studied area. Fig. 11 shows the



distribution of pressure in the entire computational domain at different times during the meteorite moving into the water column.

As shown in Fig. 11, the maximum pressure zones are observed in the frontal point of the body contact with water (in the so-called breaking point - point "1" in the figure) and under the ice surface at the start of the body immersion (point "2" in the figure).

A shock wave is observed in the fluid immediately after the meteorite entering the water. It generates a zone of high pressure around the ice surface (point "4").

Around the body the fluid accelerates from point "1" (breaking point) to point "2" (and a symmetrical point on the other side of the body). At the breaking point the pressure is maximum. As the distance from the middle of the body increases (point "2") the movement is slowed down and the pressure is reduced. In the central part of the back side of the body (near point "3") the pressure rises again, but in a very narrow area. This picture corresponds to the classic description of the boundary layer separation flow around the body with a blunt stern [Schlichting, 1960].

The picture of the pressure distribution shows the area of the possible ice destruction. It consists of two domains - a zone of high pressure in the shock wave (point "4") and a low pressure zone, which is formed after the body has passed (point "5").

In the zone of increased pressure, the possible destruction of the ice is observed when it is pushed upwards, and in the zone of reduced pressure the possible destruction will occur when it moves downwards.

## 6 Estimation of the ice cover destruction

The estimation of the ice cover destruction on Lake Chebarkul after the fall of the Chelyabinsk meteorite fragments has been recently made in [Ivanov, 2014]. The authors used both well-known empirical data of the ice destruction by explosions in the water, and the direct numerical simulation of the body impact on the ice. Both estimates, made for the meteorite vertical fall with a velocity of 100 m/s, predict that the size of the ice- hole is 6-8 m by explosive or impact action with energy 3-10 MJ, which corresponds to the energy of the Chelyabinsk meteorite.

Here we use a different approach based on the method of calculating the strength of a material [Barber, 2010]. The basis of this method is built on the assumption that the determining parameter of the structure reliability is stress or, more precisely, the state of tension at a point. The estimated value of the stress is compared with the maximum permissible stress value for the material obtained from experimental studies. The conclusion about the strength of the material is made from the comparison of its calculated and limit stresses. The strength property has a form:

$$\sigma_{max} \leq [\sigma], \tag{4}$$

where $\sigma_{max}$ is the maximum calculated stress that occurs in the material and $[\sigma]$ is the ultimate strength of the material.

Upon the impact of a meteorite, the fluid under the influence of excess pressure rushes straight up. We can consider the ice cover to be a thin elastic plate of infinite size, resting on an elastic foundation [Peschanskii, 1967] – in this case it is water. A bend occurs under this loading. In the presence of bending due to the lateral load Eq. (4) is transformed as follows:

$$M/W \leq [\sigma], \tag{5}$$





where $M$ is the bending moment due to lateral load, $W$ is the moment of resistance to bending. If the load, uniformly distributed over the area of a circle of some radius, affects infinite size ice, the maximum bending moment can be calculated to determine the load. The expression for the stress produced by the action of the maximum bending moment for winter ice at temperatures 25° C below zero is contained in [Peschanskii, 1967]:

$$q = \frac{\sigma_{max} h^{5/4}}{31\left(0.76 r_0 - h^{3/4}\right)},\tag{6}$$

where $h$ is the thickness of the ice layer, $r_0$ is the radius of the area where the load is distributed. It should be noted that this formula is obtained to determine the strength of the ice cover to allow its use as a crossing. These expressions allow us to estimate the area of lake ice cover destruction. It resulted from the under-ice pressure which arose while the body passed the ice. Given the nature of the shock pressure, one can assume that this formula allows finding an adequate estimation.

According to Fig. 11 the maximum under-ice pressure is 2 MPa, and the radius of the zone is 2 m. If the ice thickness was 1 m and the ambient temperature was about $25^0$ below zero, the stress calculated by Eq. (6) has a value of 35*103 kPa. The data of the values of river ice breaking bending stress is given in [Peschanskii, 1967]. The maximally available value is shown for the temperature of 15°C below zero and amounts to 50 kPa, which is three orders smaller than the obtained value. Therefore, it can be argued with high probability that at this point the ice cover has been destroyed. That is why the size of the ice-hole is bigger than the radius of the meteorite.

The pressure around the ice is also about 2 MPa in the reduced pressure zone. This fact means the destruction of ice in this zone. Thus, according to Fig. 9 the zone of destruction is equal to approximately 6 meters from the point of the direct fall, i.e., the overall ice-hole size obtained in the calculation, is about 7 m, which well agrees with the observed data. This estimate is also in good agreement with [Ivanov, 2014], where it is noted that an ice-hole with the diameter of about 6-7 m is to be formed in the lake if the velocity of the body-fall is about 100 m/s.

A similar method of estimating the ice cover destruction is given in [Fransson, 2009]. It suggests counti8ng the capacity of ice using the results of [Peschanskii, 1967, Barber, 2010]. Here, the force that causes the cracks in the ice is determined on the basis of the ice cover characteristic length $L$ in accordance with [Peschanskii, 1967]:

$$P_{cr} = \frac{\pi\beta}{3(1+n)f(\beta)}\sigma_f h^2,\tag{7}$$

$$f(\beta) = (0.6195 - \ln\beta)\frac{\beta}{2} + \frac{\pi\beta^2}{64} + ...,\tag{8}$$

where $\beta = r_0/L$ is the relative radius of the load; $f(\beta)$ is a function of the stress intensity; $\sigma_f$ is the bending strength of the ice cover.

The strength, leading to the ice cover destruction is determined by the semi-empirical dependence from:



$$P_u = 1.25(1+1.68\beta)\sigma_f h^2 . \tag{9}$$

Using the same data as before for the estimation, according to the Eq. (9) we find that the value of the ice cover stress is 41.3*103 kPa, which almost corresponds to the previous estimation. It is three orders greater than the ice breaking bending stress [Peschanskii, 1967]. Thus, both estimations allow explaining the observed size of the ice-hole, formed after the

meteorite entering the water, as well as the absence of ice destruction over large areas by weak tsunami waves.

## 7 Conclusions

The results of the numerical modeling of perturbations formed on a meteorite entering Lake Chebarkul, Russia are shown. The initial data corresponds to the event that occurred on February 15, 2013. The characteristics of the waves both for the water with the ice cover and without it are found. In the second case it is shown that at a distance of about 100 m from the

source the wave height is a few cm and ceases to be dangerous. In the case of a meteorite entering the lake covered with ice, the sultan which has a height of about 10 m is formed and it collapses quickly in the direction of the body motion. The main part of the sultan consists of splashes, and the height of the water bulk discharge is about 2.5 meters. The collapse zone of the sultan is about 10 m. To estimate the ice cover destruction approximate formulas are presented. They confirm the fact of breaking the ice at the site of a meteorite entering the lake. Numerical computations and estimations correctly predict the

diameter of the ice-hole, observed on the lake after a meteorite entering it, and is in good agreement with the estimates made in [Ivanov, 2014].

So, from the viewpoint of tsunami formation, the energy of the Chelyabinsk meteorite fragment that fell into the water is not enough to generate large waves. The whole effect is manifested only at the point of a meteorite entering the water. In the case of a larger meteorite fragment entering the water, tsunami waves can be significant, and estimations of danger of these

events should be made for the inland lakes and the seas.

## Acknowledgements

The presented results were obtained with the financial support of the grant of the President of the Russian Federation for state support of leading scientific schools of the Russian Federation (NSH-6637.2016.5) and RFBR grants (15-45-02061 and 16-01-00267).

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

**Figure 1: The ice-hole, formed on the surface of Lake Chebarkul.**

**Figure 2: The initial disturbance of the water surface at the source.**





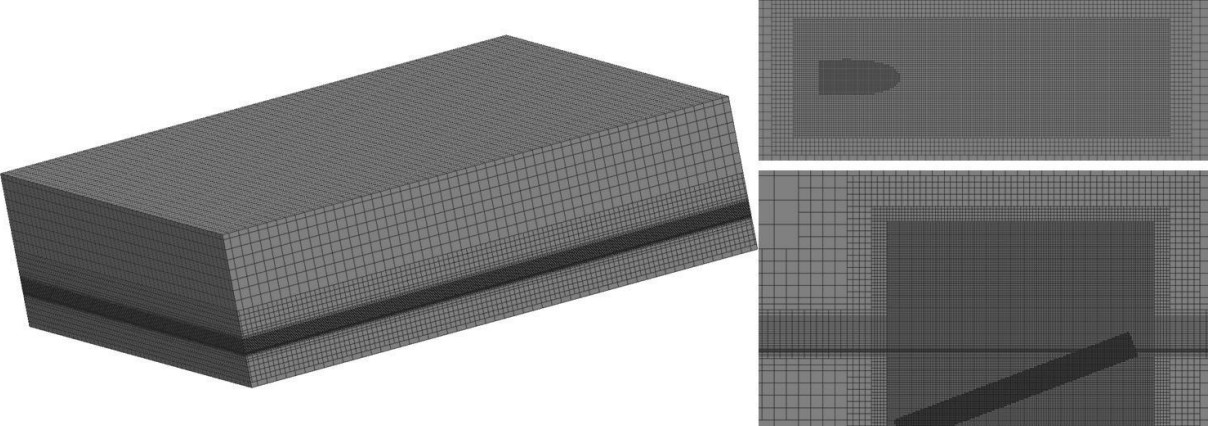

**Figure 3: Model discrete area (left) and a fragment of the computational grid at the site of the meteorite entering (right). Zones of mesh refinement near the free surface and the area of the movement of the meteorite (black).**





**Figure 4: Perturbation of water at different points in time when the meteorite enters the water at an angle of 20 degrees (the meteorite comes from left to right).**





**Figure 5: The perturbation of the volume fraction (blue - water, red - air) in the cross-section of the computational domain taken along the center line of the meteorite fall (the body enters from right to left at an angle of 20 degrees) at different times.**





**Figure 6: The form of the water surface at different times.**

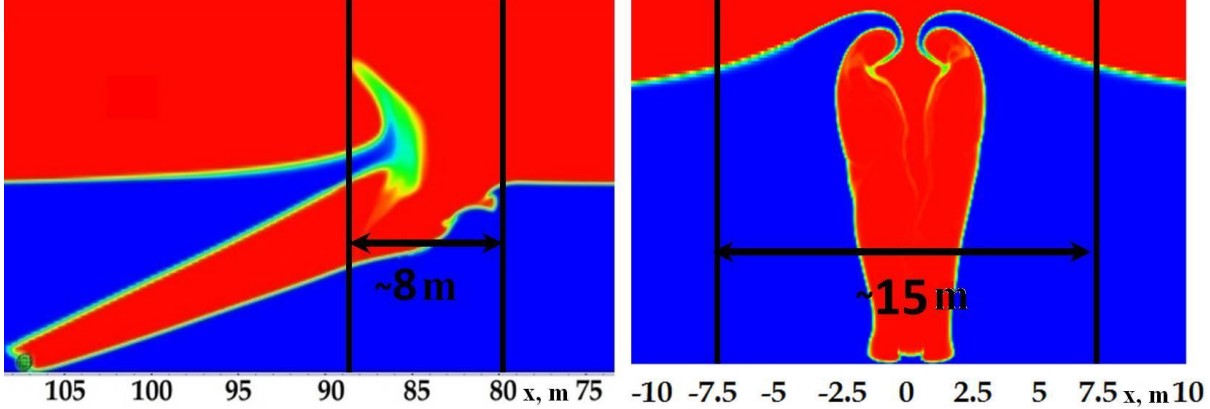

**Figure 7: The cavity formed by the meteorite fall at an angle (right) and at the vertical fall (left).**



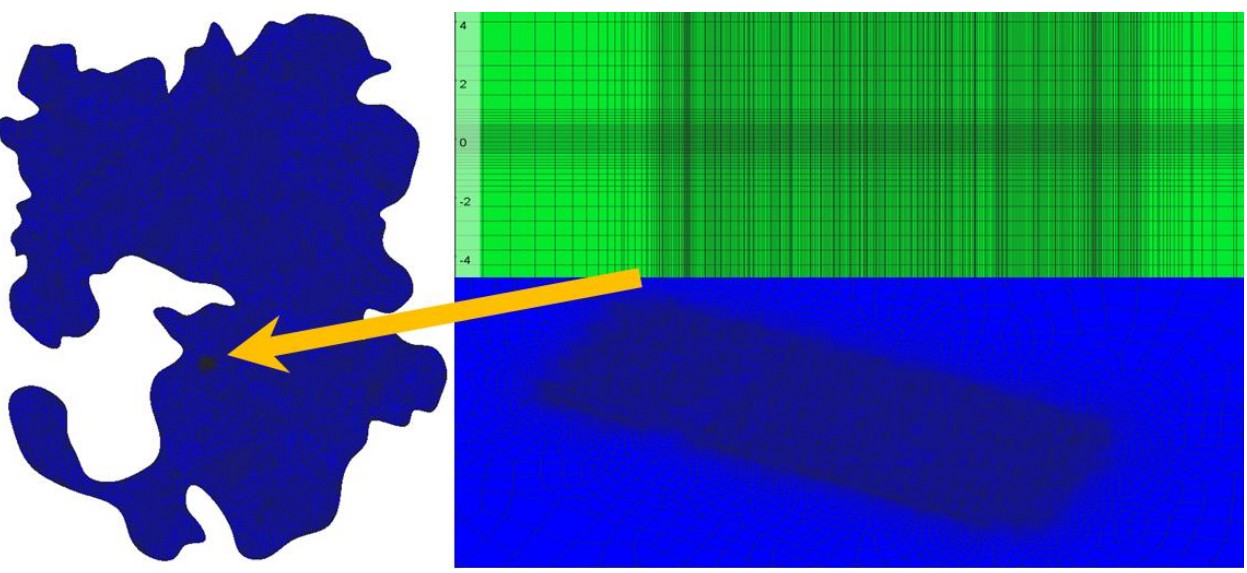

**Figure 8: The variants of the grid model for the arbitrary shore line with the marked meteorite fall zone (left – the full shore line of the lake, right – the marked fragment of the meteorite fall site).**





**Figure 9: The tsunami propagation along Lake Chebarkul.**





**Figure 10: The perturbation of the volume fraction (blue - water, white - air) in the cross-section of the computational domain taken along the center line of the meteorite fall at different times in the presence of ice (the body enters from right to left at an angle of 20 degrees).**





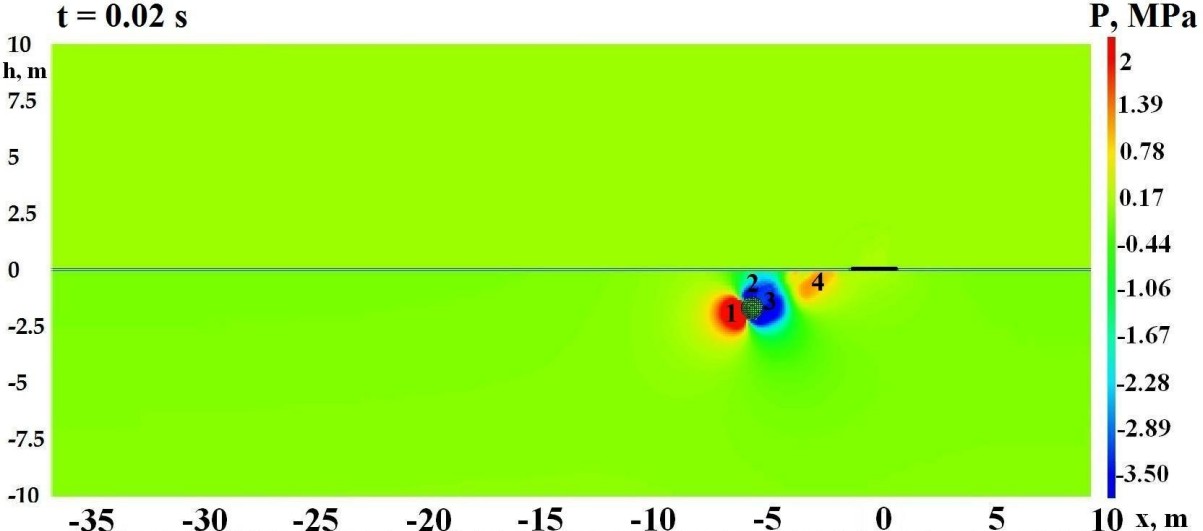

**Figure 11: The pressure field in the computational domain (the horizontal line – ice, the short black strip – entrance area).**

