# Peer review of "Numerical modeling of the 2013 meteorite entry in Chebarkul Lake, Russia"

_Natural Hazards and Earth System Sciences, 2016_

## Referee Comment (RC1) · Anonymous Referee #1 · 6 Feb 2017

Paper is focused on the numerical simulation of meteorite impact on the Chebarkul Lake in winter 2013. In simulation, the cases of lake covered by ice and lake without ice have been considered. The Navier-Stokes equations were used for two-component fluid, and numerical simulation were based on the software package LOGOS.

A details description of cavity formation, induced surface wave and ice cover destruction were given. Reviewer remarks on paper:

1. Paper is interesting for hydrodynamics community as it concerns the real case of quite recent meteorite impact. Therefore, this paper should be published in the Nat. Hazards Earth Syst. Sci. journal. However, the present form of the paper requires some improvements.

2. Paper is very difficult to read due to many unclear statements dealing with too many

details of calculations. This is particularly seen in Section 3. As the solution of the boundary-value problems was obtained using the package LOGOS, it will be sufficient to define the initial governing parameters and present the results of simulation in a more condense form.

3. Hard copies of figures is very poor visible, especially Figs. 3, 8, 9 (scattering waves for particular time steps are almost non-distinguishable).

4. Small remarks:

a) page 4, line 15: ... "bottom deformable", it means that bottom depth is not-uniform?

b) page 5, line 5: remark in the second sentence is not needed in terms of the first sentence

c) page 5, line 20: ... "meteorite covered a distance", probably it should be ... meteorite moved a distance?

d) page 7, line 21: should be Fig. 10, not Fig. 8.

---

## Referee Comment (RC2) · Anonymous Referee #2 · 7 Mar 2017

The scientific article 'Numerical modeling of the 2013 meteorite entry in Chebarkul Lake, Russia', by Andrey Kozelkov, Andrey Kurkin2, Efim Pelinovsky, Vadim Kurulin and Elena Tyatyushkina presents a study of the meteorite impact that occurred in Lake Chabarkul on the 15th of February 2013. Analysis of the meteorite entry inside a confined body of fluid has been performed by means of the Volume of Fluid method. A two-fluid system, namely air and water has been considered. The authors perform a preliminary analysis, using formulas from the relevant literature, regarding the formation of craters on the surface of the water during meteorite entry. Their analysis results to a relatively good approximation of the ice-hole diameter that was observed after the impact. Possible reasons for the discrepancies between the theoretical estimation and the actual size of the crater in the ice cover are also identified. A more detailed and accurate analysis, taking into account the mechanical behaviour of the ice-cover and

the pressure distribution acting on it, follows and more accurate simulation results are obtained.

The presented study is thorough and of good quality. The subject addressed is very interesting and within the scope of 'Natural Hazards and Earth System Sciences'. Some issues must be addressed in order to further improve the quality of the manuscript. In particular:

It is mentioned (last line, page 3) that both air and water are considered to be incompressible. Although incompressibility is a good assumption for water, air is a compressible fluid. How is the approximation of incompressibility for air justified? Is this assumption valid for the present application? Some comments on the magnitude of the Mach number might be appropriate.

In figure 4, the time instant t=0.2 sec is presented before the time instant t=0.1 sec.

In the case of the ice-covered lake, the authors use the condition 'rigid wall' to account for the ice on the surface. The ice plate has a thickness of the order of magnitude of 1m but extends for hundreds of meters. It can therefore be assigned the attributes of a slender plate. Is the 'rigid wall' approximation justified under these conditions? Flexural waves generated by the impact, propagating as hydroelastic waves, might be significant for the phenomena at the vicinity of the entry area. Several models and methods of solution for hydroelastic interactions have been proposed in the literature. It might be interesting to comment on the applicability of these models, in conjunction to the simulation strategy employed by the authors, for future studies.

Some minor typos:

Page 9, line 21 'counti8ng'

Page 2, line 15 'Also is made the estimation of the wave heights, the description of the stages of the meteorite collapse and as well as the generation of waves emanating from the source'. This sentence should be rephrased, as it is difficult to follow.

To summarise, the present scientific article is interesting and should be considered for publication as long as the above comments are addressed. A proof reading of the manuscript is also necessary in order to improve the syntax, correct typographical errors and thus maximise the potential impact of this study.

---

## Author Comment (AC1) · 14 Mar 2017

The authors would like to thank the reviewer for their thoughtful and useful comments on our paper. Below we outline how we could address specific points raised by the reviewer in a revised manuscript.

1. It is mentioned (last line, page 3) that both air and water are considered to be incompressible. Although incompressibility is a good assumption for water, air is a compressible fluid. How is the approximation of incompressibility for air justified? Is this assumption valid for the present application? Some comments on the magnitude of the Mach number might be appropriate.

The incompressibility / compressibility of air will mainly influence the rate of change of the wave parameters in the source during the collapse of the cavity. Qualitatively, the

process will be described correctly, but the quantitative difference can be significant. As rightly noted by the reviewer, this is directly related to the Mach number, determined by the characteristic velocity of the meteorite, which in our calculations is 156 m / s (line 14, page 3), which is approximately 0.5 Mach number. It is known that the compressibility of the medium must be taken into account when the Mach number exceeds 0.3. In our case, the Mach number is slightly higher than the recommended value to take into account the compressibility, and therefore, in the first stage of calculations, the compressibility can be neglected. Of course, this will introduce a certain error in the collapse of the cavern, but it is estimated that it will not exceed 15-20% for speed and pressure. That is why we did not take into account the compressibility of air in this study. After the wave emerged from the source, the effect of compressibility on the process of its propagation is negligible. We introduce this comment in text of revised paper.

2. In figure 4, the time instant t=0.2 sec is presented before the time instant t=0.1 sec.

Will be corrected in revised text.

3. In the case of the ice-covered lake, the authors use the condition 'rigid wall' to account for the ice on the surface. The ice plate has a thickness of the order of magnitude of 1m but extends for hundreds of meters. It can therefore be assigned the attributes of a slender plate. Is the 'rigid wall' approximation justified under these conditions? Flexural waves generated by the impact, propagating as hydroelastic waves, might be significant for the phenomena at the vicinity of the entry area. Several models and methods of solution for hydroelastic interactions have been proposed in the literature. It might be interesting to comment on the applicability of these models, in conjunction to the simulation strategy employed by the authors, for future studies.

To compute combine equations for water and ice requires more computer time. Taking into account that we have no enough observation data concerning this event we decided to simplify the problem and use the "rigid wall" approximation. For future studies,

of course, the "ice" block of hydroelastic equations with adequate breaking conditions should be added in the model.

4. Some minor typos:

Page 9, line 21 'counti8ng'

Corrected

Page 2, line 15 'Also is made the estimation of the wave heights, the description of the stages of the meteorite collapse and as well as the generation of waves emanating from the source'. This sentence should be rephrased, as it is difficult to follow.

We deleted this sentence.

---

## Author Comment (AC2) · 14 Mar 2017

The authors would like to thank the reviewer for their thoughtful and useful comments on our paper. Below we outline how we could address specific points raised by the reviewer in a revised manuscript.

1. Paper is very difficult to read due to many unclear statements dealing with too many details of calculations. This is particularly seen in Section 3. As the solution of the boundary-value problems was obtained using the package LOGOS, it will be sufficient to define the initial governing parameters and present the results of simulation in a more condense form.

We changed this section. We will keep here the initial governing parameters in revised form of our manuscript. Most of details will be moved from Section 3 to Appendix

because they can be important for modelers.

2. Hard copies of figures is very poor visible, especially Figs. 3, 8, 9 (scattering waves for particular time steps are almost non-distinguishable).

The quality of figures is improved due to higher resolutions.

4. Small remarks: a) page 4, line 15: ... "bottom deformable", it means that bottom depth is not-uniform?

Sorry, it is mistake. The bottom is non-deformable

b) page 5, line 5: remark in the second sentence is not needed in terms of the first sentence

This sentence will be reformulated.

c) page 5, line 20: ... "meteorite covered a distance", probably it should be ... meteorite moved a distance?

Sorry for mistake. It is corrected on: meteorite has passed a distance of about 27 meters

d) page 7, line 21: should be Fig. 10, not Fig. 8.

It will be corrected.

---

## Author Response (AR1)

The authors would like to thank the reviewers for their thoughtful and useful comments on our paper. Below we outline how we could address specific points raised by the reviewer in a revised manuscript.

**Review 1**

*1. Paper is very difficult to read due to many unclear statements dealing with too many details of calculations. This is particularly seen in Section 3. As the solution of the boundary-value problems was obtained using the package LOGOS, it will be sufficient to define the initial governing parameters and present the results of simulation in a more condense form.*

We changed Section 3.
Page 4, line 33: "The bathymetric data of Lake Chebarkul are not available, and the depth of a model region relative to the surface is taken constant of 10 m, corresponding to an average depth of Lake Chebarkul. The height above the zero level surface of the water is 40 m. So, the model discrete area with the size of 160×160×50 m was constructed. It is a non-structured three-dimensional (3D) grid of truncated polyhedrons of arbitrary shape (Fig. 3). This type of grid is the only possible one for the areas of complex geometric configurations. It is built by the preprocessor of the software package LOGOS."

*2. Hard copies of figures is very poor visible, especially Figs. 3, 8, 9 (scattering waves for particular time steps are almost non-distinguishable).*

The quality of figures is improved due to higher resolutions.

*4. Small remarks:*
*a) page 4, line 15: ... "bottom deformable", it means that bottom depth is not-uniform?*

Page 4, line 23: "non-deformable"

*b) page 5, line 5: remark in the second sentence is not needed in terms of the first sentence*

Page 5, line 5: "In the first numerical experiment within the hydrodynamic model described above, we examine the case of wave generation in the water without ice cover."

*c) page 5, line 20: ... "meteorite covered a distance", probably it should be ... meteorite moved a distance?*

Page 5, line 19: "it has passed a distance of about 27 meters"

*d) page 7, line 21: should be Fig. 10, not Fig. 8.*

Page 7, line 21: "Fig. 10"

**Review 2**

*1. It is mentioned (last line, page 3) that both air and water are considered to be incompressible. Although incompressibility is a good assumption for water, air is a compressible fluid. How is the approximation of incompressibility for air justified? Is this assumption valid for the present application? Some comments on the magnitude of the Mach number might be appropriate.*

Page 3, line 25: "The incompressibility / compressibility of air will mainly influence the rate of change of the wave parameters in the source during the collapse of the cavity. Qualitatively, the process will be described correctly, but the quantitative difference can be significant. This is directly related to the Mach number, determined by the characteristic velocity of the meteorite,

which in our calculations is 156 m/s, which is approximately 0.5 Mach number. It is known that the compressibility of the medium must be taken into account when the Mach number exceeds 0.3. In our case, the Mach number is slightly higher than the recommended value to take into account the compressibility, and therefore, in the first stage of calculations, the compressibility can be neglected. Of course, this will introduce a certain error in the collapse of the cavern, but it is estimated that it will not exceed 15-20% for speed and pressure. After the wave emerged from the source, the effect of compressibility on the process of its propagation is negligible."

*2. In figure 4, the time instant t=0.2 sec is presented before the time instant t=0.1 sec.*

Page 16: In figure 4, now the time instant t = 0.2 sec is presented after the time instant t = 0.1 sec.

*3. In the case of the ice-covered lake, the authors use the condition 'rigid wall' to account for the ice on the surface. The ice plate has a thickness of the order of magnitude of 1m but extends for hundreds of meters. It can therefore be assigned the attributes of a slender plate. Is the 'rigid wall' approximation justified under these conditions? Flexural waves generated by the impact, propagating as hydroelastic waves, might be significant for the phenomena at the vicinity of the entry area. Several models and methods of solution for hydroelastic interactions have been proposed in the literature. It might be interesting to comment on the applicability of these models, in conjunction to the simulation strategy employed by the authors, for future studies.*

Page 4, line 15: "To compute combine equations for water and ice requires more computer time. Taking into account that we have no enough observation data concerning this event we simplified the problem and use the "rigid wall" approximation to account for the ice on the surface, all boundaries and the bottom of the computational domain (distant from the site of a meteorite entering the water). In the present calculations zero initial conditions for the velocity field in the water, the water level and the standard hydrostatic distribution of the pressure are used also."

Page 4, line 27: "For future studies, of course, the "ice" block of hydroelastic equations with adequate breaking conditions should be added in the model."

*4. Some minor typos:*

*Page 9, line 21 'counti8ng'*

Page 9, line 21: "counting"

[revised manuscript text omitted]

---

## Referee Report (RR1)

The updated version of the scientific article 'Numerical modeling of the 2013 meteorite entry in Chebarkul Lake, Russia', by Andrey Kozelkov, Andrey Kurkin2, Efim Pelinovsky, Vadim Kurulin and Elena Tyatyushkina is significantly improved both in terms of content and presentation style. However, some technical corrections, related mostly to the use of English language, are still needed in order to improve the syntax and thus the final result.

For example,

Page 2, line 10:

'First of all, in section 2 is used a well-known parametric model …', should become 'First of all, in section 2 a well-known parametric model … is used.'

Page 4, line 15

'To compute combine equations for water and ice requires more computer time.'
This sentence e.g. could be written as:
'The numerical solution of coupled, water wave/ice deformation differential equations is computationally intensive…'.

It is this reviewer's suggestion that the authors implement corrections and adjustments similar to the above throughout the manuscript in order to improve the syntax and maximise the impact of the manuscript.

---

## Author Response (AR2)

The authors would like to thank the reviewers for their thoughtful and useful comments on our paper. Below we outline how we could address specific points raised by the reviewer in a revised manuscript.

**Review**

*The updated version of the scientific article 'Numerical modeling of the 2013 meteorite entry in Chebarkul Lake, Russia', by Andrey Kozelkov, Andrey Kurkin, Efim Pelinovsky, Vadim Kurulin and Elena Tyatyushkina is significantly improved both in terms of content and presentation style. However, some technical corrections, related mostly to the use of English language, are still needed in order to improve the syntax and thus the final result.*

We have made some technical corrections, related to the use of English language.

Page 2, line 10: "First of all, in section 2 a well-known parametric model of the tsunami source of meteoric origin [Ward and Asphaug, 2000] which estimates the disturbances in the water at the site of the meteorite entry is used."

Page 2, line 17: "In section 6 the results of estimations of the ice cover destruction, based on the calculation of the stresses, are shown."

Page 3, line 15: "The numerical solution of coupled, water wave/ice deformation differential equations is computationally intensive. Due to absence of observations related to waves induced by this event, there is no possibility to refine the calculations, and for the first step the simplified problem can be considered with "rigid wall" boundary conditions at all spatial boundaries of the computational domain."

Page 5, line 24: "At the time of the cavity collapse a "primary wave" with the height of about 3.5 meters and large air pocket in the water column are generated."

Page 5, line 30: "At this time, the "secondary" wave of almost the same height as the primary one is generated."

Page 6, line 24: "As it was already noted, the bathymetric map of Lake Chebarkul is not available. It is possible to examine tsunami wave propagation on the lake using digitized coastline and constant average depth (for lakes Chebarkul it is 10 meters). The map of Lake Chebarkul shoreline and its digitized version, built with model grid (right), are shown in Fig. 8. As we can see, the coastline has a complicated configuration, and it is advisable to construct a model grid with the help of an automatic generator of arbitrary unstructured grids."

Page 6, line 29: "The grid constructed by this generator is shown in Fig. 8 (left). In the area free from the shoreline (open water) the grid has is mainly a hexagonal structure. Lake dimensions in length and width are approximately the same and are about 2 km. With the meteorite diameter of 1 meter it is reasonable to allocate a fall area to grind the model grid to the desired level (Fig. 8, right) – at least 10 cells along the meteorite diameter (typical cell size of 10 cm)."

Page 7, line 2: "To simulate the wave propagation accurately it is necessary to thicken the grid near the interface of air and water surface (Fig. 8, top right). Usually, this thickening is carried out by the given law of geometric progression, so the typical cell sizes near the surface can be selected of the desired size in order to track the desired wave height. The grid consists of approximately 15 million cells."

Page 7, line 7: "Already at the 9[th] second we can see that the wave has sufficiently small amplitude of about 10 cm. About 30 seconds after the fall the wave reaches the nearest shore, and its height is also about 10 cm. At the 45[th] second we can see the bounce of the wave from the shore. This, in its turn, suggests a fairly good grid resolution which allows to reproduce centimeter amplitude waves. At the 60[th] second we can see how the bounced wave follows the primary one, which arose after the collapse of the water crater, formed in the course of a meteorite fall."

[revised manuscript text omitted]